# Mitotic Dysregulation at Tumor Initiation Creates a Therapeutic Vulnerability to Combination Anti-Mitotic and Pro-Apoptotic Agents for MYCN-Driven Neuroblastoma

**DOI:** 10.3390/ijms242115571

**Published:** 2023-10-25

**Authors:** Lei Zhai, Anushree Balachandran, Rebecca Larkin, Janith A. Seneviratne, Sylvia A. Chung, Amit Lalwani, Shoma Tsubota, Dominik Beck, Kenji Kadomatsu, Anneleen Beckers, Kaat Durink, Katleen De Preter, Frank Speleman, Michelle Haber, Murray D. Norris, Alexander Swarbrick, Belamy B. Cheung, Glenn M. Marshall, Daniel R. Carter

**Affiliations:** 1Children’s Cancer Institute Australia, Lowy Cancer Research Centre, University of New South Wales, Randwick, NSW 2031, Australia; 2Adult Cancer Program, Lowy Cancer Research Centre, University of New South Wales, Sydney, NSW 2031, Australia; 3Department of Biochemistry, Nagoya University Graduate School of Medicine, Nagoya University, Nagoya 466-8550, Japan; 4School of Biomedical Engineering, University of Technology Sydney, Ultimo, NSW 2007, Australia; 5Department of Biomolecular Medicine, Cancer Research Institute Ghent, Ghent University, 9000 Ghent, Belgium; 6UNSW Centre for Childhood Cancer Research, University of New South Wales, Sydney, NSW 2031, Australia; 7The Kinghorn Cancer Centre, Garvan Institute of Medical Research, Darlinghurst, NSW 2010, Australia; 8School of Women’s and Children’s Health, University of New South Wales, Randwick, NSW 2031, Australia; 9Kids Cancer Centre, Sydney Children’s Hospital, Randwick, NSW 2031, Australia

**Keywords:** neuroblastoma, tumorigenesis, MYCN, mitosis, combination therapy

## Abstract

*MYCN* amplification occurs in approximately 20–30% of neuroblastoma patients and correlates with poor prognosis. The *TH-MYCN* transgenic mouse model mimics the development of human high-risk neuroblastoma and provides strong evidence for the oncogenic function of MYCN. In this study, we identified mitotic dysregulation as a hallmark of tumor initiation in the pre-cancerous ganglia from *TH-MYCN* mice that persists through tumor progression. Single-cell quantitative-PCR of coeliac ganglia from 10-day-old *TH-MYCN* mice revealed overexpression of mitotic genes in a subpopulation of premalignant neuroblasts at a level similar to single cells derived from established tumors. Prophylactic treatment using antimitotic agents barasertib and vincristine significantly delayed the onset of tumor formation, reduced pre-malignant neuroblast hyperplasia, and prolonged survival in *TH-MYCN* mice. Analysis of human neuroblastoma tumor cohorts showed a strong correlation between dysregulated mitosis and features of *MYCN* amplification, such as MYC(N) transcriptional activity, poor overall survival, and other clinical predictors of aggressive disease. To explore the therapeutic potential of targeting mitotic dysregulation, we showed that genetic and chemical inhibition of mitosis led to selective cell death in neuroblastoma cell lines with MYCN over-expression. Moreover, combination therapy with antimitotic compounds and BCL2 inhibitors exploited mitotic stress induced by antimitotics and was synergistically toxic to neuroblastoma cell lines. These results collectively suggest that mitotic dysregulation is a key component of tumorigenesis in early neuroblasts, which can be inhibited by the combination of antimitotic compounds and pro-apoptotic compounds in MYCN-driven neuroblastoma.

## 1. Introduction

Neuroblastoma is a childhood cancer of the sympathetic nervous system that commonly arises in the abdomen, either from the adrenal medulla or sympathetic ganglia [1]. Due to the developmental origin of the disease, it almost exclusively occurs in infancy and early childhood [2]. While some cases of neuroblastoma undergo spontaneous or post-therapy regression, clinically advanced neuroblastoma, which presents in 50% of patients, is often therapy-resistant or relapses after an initial response to treatment [3].

Clinical and pathological features, such as age at diagnosis, stage, and tumor histology are used to diagnose and classify high-risk patients. At the molecular level, *MYCN* amplification independently indicates high-risk neuroblastoma that is associated with poor prognosis [1]. MYCN regulates early sympathoadrenal development in the developing fetus and alterations to its expression are associated with neuroblastoma [2,4]. Consistent with this, human *MYCN* transgene expression driven by rat *tyrosine hydroxylase* (*TH*) promoter in the transgenic *TH-MYCN* mice, is sufficient to recapitulate many of the features of human neuroblastoma [5].

The timeline of tumorigenesis in *TH-MYCN* mice has been previously characterized [6,7,8]. Neuroblastoma tumor formation in this model is preceded by hyperplasia of premalignant neuroblasts in the paravertebral ganglia of the neck, thorax, and abdomen in the first weeks after birth. This hyperplasia is not a determinant of tumor development and in most cases, only abdominal coeliac ganglia progress to form tumors, while other hyperplastic ganglia of the neck and thorax only rarely progress to tumors. In comparison, wild-type mice display neuroblast hyperplasia in paravertebral ganglia at birth that completely regresses by 2 weeks of age [8]. This suggests that there are additional requirements that allow neuroblast hyperplasia, persistence, and transformation to ultimately allow tumor development [2]. However, the mechanism that promotes tumor initiation instead of regression in these neuroblasts is yet unknown [2,9]. Studying the paravertebral ganglia during the precancerous phase in *TH-MYCN* mice offers a valuable model to unravel the molecular mechanisms of tumor initiation in MYCN-driven neuroblastoma [6,7,8,10,11,12,13,14,15].

In this study, bulk transcriptomics and comparative single-cell qPCR analysis of premalignant ganglia and tumors from *TH-MYCN^+/+^* mice showed the acquisition of mitotic dysregulation through tumorigenesis. Prophylactic treatment with antimitotic compounds barasertib and vincristine reduced tumor incidence and prolonged survival in homozygous *TH-MYCN^+/+^* mice. Additionally, combination treatment using an antimitotic and pro-apoptotic compound was synergistically toxic to human neuroblastoma cell lines and *TH-MYCN^+/+^* tumors. Collectively our results suggest that mitotic dysregulation is a key feature that contributes early to neuroblastoma tumor initiation and that a combination of antimitotic compounds and pro-apoptotic compounds offers synergistic therapeutic benefit in MYCN-driven neuroblastoma.

## 2. Results

### 2.1. Gene Expression Profiling of TH-MYCN^+/+^ Transgenic Mice Identifies Mitotic Gene Set Enrichment in Pre-Malignant Neuroblasts

To evaluate the transcriptome of MYCN-driven neuroblastoma through the course of tumor initiation and progression, we compared bulk mRNA expression profiles of ganglion tissues derived from *TH-MYCN^+/+^* 1 and 2-week-old mice, and tumors from 6-week-old mice, to that of age-matched tissue from wild-type littermates [11]. We used a linear predictor model [15] to identify genes whose expression diverged between wild-type and *TH-MYCN^+/+^* mice over time (Figure 1A). This identified three gene clusters with distinct expression patterns, in particular a group defined as “Gene Group 3” with gene expression patterns that indicate progressive upregulation in *TH-MYCN^+/+^* ganglia and progressive downregulation in wildtype ganglia (Figure 1B). Gene Ontology (GO) term over-representation analysis for these three gene groups identified unique pathway enrichment, with Gene Group 3 having a strong mitotic gene enrichment including GO terms such as “spindle assembly” and “mitotic nuclear division” amongst others (Figure 1C, Appendix A). Examination of Gene Group 3 identified numerous mitotic regulators, and indeed a 9 gene mitotic gene signature (MGS) strongly represented the divergent expression patterns seen when comparing wildtype and *TH-MYCN^+/+^* tissues (Figure 1A, see red text, Figure 1D). The MGS was strongly correlated with a target gene signature for the MYC oncogene [16] (Figure 1E). Since this MYC signature is known to be highly concordant with MYCN transcriptional activity [7], this suggests a regulatory link between the MGS and MYCN in TH-MYCN^+/+^ ganglia.

A hallmark of neuroblastoma initiation in *TH-MYCN^+/+^* mice is the presence of increased numbers of premalignant hyperplastic neuroblasts in sympathetic ganglia [8]. To investigate MGS expression in these cells, we performed targeted single-cell qPCR on premalignant ganglia cells from *TH-MYCN^+/+^* mice at day 10 and tumor cells from 40-day-old *TH-MYCN^+/+^* tumors (Figure 1F). To classify cell types, we quantified the expression of marker genes for cell types expected in sympathetic ganglia [7,17,18,19] including neuroblasts (*Phox2b*, *hMYCN*, *Dlk1*), ganglion cells (*Gap43*, *Dbh*, *Tubb3*) and Schwannian cells (*Sox10*, *Col1a2*, *S100a1*) (Figure 1F). Comparing the expression of the 9 genes of the MGS, showed significantly higher expression in the neuroblasts compared with other cell types (Figure 1F,G). Interestingly however, the neuroblasts of 10-day-old ganglia, and 40-day-old tumor tissue showed no significant difference in MGS expression patterns, suggesting that mitotic dysregulation occurs as an early feature of tumor development in hyperplastic neuroblasts (Figure 1G). This specific relationship was evident where MGS expression was correlated with markers of immature neuroblasts such as *Phox2b* and indeed *hMYCN*, the driver oncogene of this model (Figure 1H, Appendix A). Together, these results suggest that MYCN-driven tumor initiation is accompanied by early and marked upregulation of the MGS, and mitotic dysregulation could be the pathway by which MYCN shifts the premalignant neuroblast toward malignant transformation.

### 2.2. Prophylactic Inhibition of Mitosis Impedes MYCN-Driven Tumor Initiation and Progression to Neuroblastoma

To investigate the functional role of mitotic dysregulation in pre-malignant neuroblasts, we treated *TH-MYCN^+/+^* mice prophylactically with either barasertib, a small molecule inhibitor of AURKB (regulates alignment and segregation of chromosomes during mitosis) or vincristine, standard antimitotic chemotherapy against neuroblastoma that inhibits microtubule formation in the mitotic spindle [1,20]. Six-day-old *TH-MYCN^+/+^* mice were treated with intraperitoneal vehicle controls, vincristine (0.05 mg/kg/day), or barasertib (25 mg/kg/day) for 4 days per week over 5 weeks, and thereafter assessed for tumor growth. Treatment with anti-mitotic compounds significantly improved survival compared to vehicle controls (Figure 2A). Tumor growth was completely inhibited in 40% of barasertib- and 60% of vincristine-treated mice up to 210 days. This pronounced effect occurred despite doses administered at a concentration 2–4 fold lower than previously reported maximum-tolerated doses in mice [7,20] and no adverse effect was observed on body weight (Appendix A). We further quantified the effect of either barasertib or vincristine treatment given from postnatal day 6–9, on the level of neuroblast hyperplasia in the coeliac ganglia at 12 days of age. In comparison to vehicle controls, we found a significant 3.1–4.3 fold reduction in the size of hyperplastic regions in treated mice, compared with vehicle control (Figure 2B,C). These data show that mitotic dysregulation during early MYCN-driven tumor initiation is required for rapid progression to tumor formation.

Since mitotic dysregulation is known to prime cells for apoptotic cell death [21,22], we hypothesized that this functional link would exist in vivo at the stages of tumor initiation and progression. To test this, we administered the pro-apoptotic compound ABT263, a pan-BCL2 family inhibitor [23], to *TH-MYCN^+/+^* mice prophylactically from day 6 for 5 weeks. Similar to antimitotic compounds, the Kaplan–Meier analysis revealed that ABT263-treated mice showed a significantly prolonged survival and decreased tumor incidence when compared to vehicle-treated mice (Appendix A). Histological analysis was performed in the coeliac ganglia of 12-day-old *TH-MYCN^+/+^* mice and revealed a marked decrease in the area of hyperplasia following 4 doses of ABT263 (Appendix A). These findings confirm a role for anti-apoptotic BCL2 proteins in *TH-MYCN* tumor initiation and point to a functional link between mitotic dysregulation and the anti-apoptotic machinery.

### 2.3. High Expression of Mitotic Genes Is Associated with Poor Prognosis in Neuroblastoma Patients

We next examined microarray expression data from a large primary, treatment-naïve, human neuroblastoma cohort (Kocak cohort, n = 475; GEO GSE45480) [24]. We classified patient tumors based on the expression of the previously identified genes from the *TH-MYCN* ganglia analysis in Figure 1A. Hierarchical clustering separated the tumor cohort into three groups, with MGS expression presenting as the dominant signal defining these tumors (Figure 3A, see red text, Figure 3B). Tumors classified with a high MGS score (MGS.High) showed significantly worse overall survival compared with tumors defined with intermediate or low expression of the MGS (MGS.Int or MGS.Low respectively) (Figure 3C). Similar to *TH-MYCN* ganglia, MGS expression also correlated with tumors that had a high MYC or MYCN transcriptional activity (Figure 3D) and MGS expression was significantly higher in clinical categories of poor prognosis for *MYCN* amplification, metastasis (International neuroblastoma staging system—INSS4) and older age (>18 months) compared with their lower risk counterparts (Figure 3E). Similar results were observed in a separate cohort based on RNA-seq data (Appendix A) [25].

To assess the functional requirement of mitotic genes in established neuroblastoma, we next quantified the effect of silencing two representative genes within the MSG in neuroblastoma cell lines, BUB1 and KIFC1. We also targeted AURKB, a mitotic gene that has been previously shown to have a MYCN signal dependency [20]. We chose cell lines that represent a range of hallmark features of neuroblastoma: MYCN-amplified (SK-N-BE(2)C) vs. non-amplified (SH-EP + SK-N-AS), as well as cell lines conforming to the adrenergic (SK-N-BE(2)C), mixed (SK-N-AS) and mesenchymal (SH-EP) super-enhancer classification [26,27,28]. We first validated that siRNA-mediated knockdown of these genes inhibited protein expression of the corresponding genes in three neuroblastoma cell lines, SK-N-BE(2)C, SH-EP, SK-N-AS (Figure 3F). Following gene knockdown, the percentage of viable cells became progressively lower through 48-, 72-, and 96-h post-transfection, with later timepoints shown to be significantly lower compared with cells transfected with non-targeting siRNA (Figure 3G). Moreover, a marked reduction in colony formation was also observed in neuroblastoma cell lines upon mitotic gene knockdown (Figure 3H,I) suggesting a dependence on mitotic genes for cell survival and/or proliferation. Interestingly, across cell viability and colony assays, mitotic genes show similar potency regardless of MYCN amplification status or super-enhancer classification status of cell lines. To determine whether *AURKB*, *BUB1,* and *KIF1C* are MYCN target genes, we performed ChIP-PCR and demonstrated a 4-to-10-fold enrichment of MYCN binding to the promoter regions in two *MYCN*-amplified neuroblastoma cell lines (Figure 3J). Moreover, MYCN ChIP sequencing in MYCN-amplified cell lines [29], shows evidence of MYCN transcriptional regulation at the promoter region of all MGS members (Appendix A). These data indicate that MGS expression in established neuroblastoma is strongly linked to poor prognosis, MYCN transcriptional activity, and clinical predictors of aggressive disease. Moreover, mitotic gene overexpression is essential to neuroblastoma cell viability and clonogenicity in neuroblastoma cells.

### 2.4. MYCN Overexpression Sensitizes Neuroblastoma Cells to Mitotic Gene Knockdown, by Induction of Apoptosis

To investigate the effect of MYCN on mitotic dysregulation and its functional consequences in neuroblastoma cells, we performed siRNA knockdown of AURKB, BUB1, and KIFC1 in MYCN-regulable SHEP21N cells (Figure 4A). This showed conditional cytotoxicity after 72–96 h, where siRNA for all genes reduced cell viability in MYCN-overexpressing SHEP21N cells to a greater extent than cells lacking MYCN expression (Figure 4B). As apoptosis has been reported to be a functional consequence of MYC-driven mitotic dysregulation [20,21,22], we compared the extent of apoptosis in SHEP21N cells transiently transfected with AURKB, BUB1, and KIFC1 siRNA. Annexin V/7-AAD assay showed there was an increase in the early and late apoptotic population after mitotic gene knockdown, with a trend that MYCN-expressing cells were undergoing early apoptosis (Figure 4C and Appendix A). To validate this, we explored the expression of cleaved PARP, a marker of intrinsic apoptosis. This showed a marked increase for cleaved PARP in the MYCN-expressing SHEP21N cells following siRNA-mediated knockdown of mitotic genes (Figure 4D). These findings suggest that mitotic gene knockdown induces selective cytotoxicity and apoptosis in the presence of MYCN.

### 2.5. Chemical Inhibition of Mitosis Is Selectively Toxic to MYCN-Expressing Neuroblastoma

To quantify the toxicity of anti-mitotic compounds in the presence of MYCN, we treated MYCN-regulable SHEP21N cells with barasertib, vincristine, and VX-680, a pan-aurora kinase inhibitor. All three compounds exhibited significantly higher cytotoxicity when MYCN expression was induced (Figure 5A). Upon conducting cell cycle analysis with propidium iodide staining, we observed an increase in the percentage of cells in the sub G1 phase and also the induction of polyploidy in barasertib and VX-680 in the MYCN-overexpressing cells (Figure 5B, Appendix A). Similar to our mitotic gene-specific siRNA experiments above, these results suggested induction of apoptosis, which was confirmed using cleaved PARP western blotting and the Annexin V/7-AAD assay (Figure 5C, Appendix A).

### 2.6. Combination Antimitotic/Pro-Apoptotic Therapy Is Efficacious in MYCN-Driven Neuroblastoma

Prophylactic administration of both antimitotic and pro-apoptotic compounds was efficacious in preventing tumor progression in *TH-MYCN* mice (Figure 2, Appendix A). To investigate the combined antimitotic/pro-apoptotic efficacy in established MYCN-driven neuroblastoma as a therapeutic option, we treated SHEP21N with the combination of antimitotic compounds and pro-apoptotic compounds at various dosages in a MYCN-negative and MYCN-induced setting (Figure 6A). We used three antimitotic compounds (barasertib, vincristine, and VX-680) and two targeted pro-apoptotic compounds (ABT199 and S63845). ABT199 is a selective BCL2 inhibitor that can be delivered orally and has fewer side effects than ABT263 in vivo [30,31]. S63845 is a selective MCL-1 inhibitor that does not bind to other members in the BCL2 family [32]. Significant synergy was observed in 4/6 combinations when compared to theoretical BLISS combination additivity (Figure 6A–C). Notably, the potency of the combination therapies was more pronounced in the MYCN-expressing setting for all compounds (Figure 6A–C). Based on these criteria, we selected the S63845 + barasertib combination as the most promising, since it had the best selectivity for MYCN in terms of combination synergism and potency (Figure 6C). Accordingly, combination efficacy extended to an in vivo setting of *TH-MYCN^+/+^* mice with established tumors, where S63845 + barasertib outperformed the vehicle control or single agents in terms of tumor growth delay (Figure 6D). These data collectively show that therapeutic strategies that exploit the selective relationships between mitotic, anti-apoptotic, and the *MYCN* oncogene show promise for improved neuroblastoma combination therapy treatment.

## 3. Discussion

The key molecular steps that support MYCN-driven tumor initiation are not entirely resolved. Here we identify early and marked mitotic dysregulation in pre-tumor neuroblasts of the *TH-MYCN^+/+^* mouse model of neuroblastoma. MYCN acts as a transcriptional regulator of mitotic genes and leads to subsequent upregulation and mitotic dysregulation. Prophylactic treatment of *TH-MYCN^+/+^* mice with chemical inhibitors of mitosis was sufficient to largely block further tumor progression, suggesting an MYCN-specific molecular vulnerability. Indeed, this MYCN dependence on mitotic dysregulation was seen in cell lines of established neuroblastoma, where genetic or chemical inhibitors of mitosis were selectively toxic when MYCN was over-expressed, leading to the induction of apoptosis. Finally, we demonstrated that an antimitotic/pro-apoptotic targeted combination therapy was synergistic against MYCN, making them efficacious in cell lines and animal models of neuroblastoma.

*TH-MYCN^+/+^* mice have proved to faithfully recapitulate tumorigenesis similar to the human disease [5]. A multitude of molecular links have been identified that support the development of tumors, many of which likely converge on the regulation of mitosis [2,6,7,8,10,11,13,14,15,34,35]. Our analysis showed that mitotic dysregulation occurred early in this process. Based on mitotic gene expression levels in single-cell qPCR, this indicated that premalignant neuroblasts have a similar extent of mitotic dysregulation as tumor cells in *TH-MYCN^+/+^* mice. The functional requirement for mitotic dysregulation in tumor initiation is likely associated with the hyperplastic phenotype seen in pre-malignant sympathetic ganglia, providing a proliferative advantage. However, since hyperplasia is a feature of extra-abdominal sympathetic ganglia that rarely progresses to tumor formation in *TH-MYCN^+/+^* mice [5,8], there must be an additional requirement for cell transformation and subsequent tumor progression. A possibility is that mitotic dysregulation links to genomic instability [22], which may be favorable for clonal selection and genetic evolution of transforming neuroblasts. Interestingly, the anti-BCL2 family inhibitor ABT263, was similarly potent in preventing tumor progression in *TH-MYCN^+/+^* mice, a finding that is reminiscent of MYC-driven lymphoma models [36]. This suggests mitotic dysregulation could be accompanied by mitotic stress and genomic instability, making cells vulnerable to apoptosis when exposed to chemical inhibitors [21,22]. Dependency on mitotic/anti-apoptotic dysregulation in *TH-MYCN^+/+^* tumorigenesis is further supported since other chemical agents used for *TH-MYCN^+/+^* prophylaxis were less effective at tumor prevention, for instance, genotoxic or metabolic targeting agents [7,34]. Further genetic analyses of *TH-MYCN^+/+^* neuroblasts during early initiation will unravel potential genetic evolution mechanisms underlying tumorigenesis. Moreover, research into this narrow window in which cells are vulnerable to mitotic inhibition may support future cancer prevention or early detection therapeutics that exploit this conditional sensitivity.

Our study also identified that mitotic dysregulation was highly correlated with MYCN activity in established neuroblastoma. In a large tumor cohort of neuroblastoma [24], tumors with high mitotic gene expression were identified as being highly aggressive with expression of MYC target genes, likely stemming from *MYCN* amplification or high cMYC activity [7,37]. We also showed that siRNA knockdown of mitotic genes in established neuroblastoma cell lines was accompanied by reduced cell viability and colony-forming potential, irrespective of cell line *MYCN*-amplification status. This suggests that while mitotic genes are essential for cell viability/proliferation to some degree in all neuroblastoma cells, there may be a molecular dependency on MYCN. Indeed, we showed this in a MYCN-inducible cell model, where MYCN overexpression made cells more sensitive to genetic and chemical inhibition of mitosis. This occurred by induction of a pro-apoptotic phenotype, suggesting MYCN and mitotic dysregulation primes cells for apoptosis but cancer-specific cell alterations allow survival, as has been demonstrated previously [21,22].

Our therapeutic investigation further pursued an antimitotic/pro-apoptotic strategy, showing a MYCN-specific potentiation of certain combination therapies. Using an MYCN-inducible model, we showed that barasertib (AURKB inhibitor) plus S63845 (MCL1 inhibitor) showed the best MYCN selectivity in terms of synergism and potency, to our knowledge the first example of this approach. Moreover, barasertib/S63845 combination therapy significantly prolonged the survival of *TH-MYCN^+/+^* mice with established tumors. These data suggest that there is potential for new combination therapies targeting mitosis and anti-apoptotic machinery, especially in light of disappointing clinical findings using single-agent antimitotic strategies in multiple cancers [38]. Since neuroblastoma demonstrates some potential for sensitivity to antimitotic approaches in the clinic such as AURKA inhibitors [39], this suggests further investigation is warranted for the molecular interactions between MYCN and mitosis, in particular those that represent potential vulnerabilities to new combination therapies [40,41].

This study includes some limitations. While we have undertaken correlative analysis on human tumors linking MYCN and mitosis, our experimental findings are primarily obtained by using artificial models, such as isogenic cell lines and genetically engineered mouse models. While these models are useful for controlling confounding variables, future studies would benefit from testing more natural cancer models, for example, comparing MYCN amplified cell lines with MYCN non-amplified cell lines or patient-derived xenografts, which might serve as useful tools for assessing MYCN links to mitosis. One challenge will be to determine if this effect is recapitulated in other cancers with recurrent MYCN amplification such as medulloblastoma and neuroendocrine prostate cancer or in the numerous cancers in which MYC is altered. It is interesting that adult cancers with MYC alterations show similar observations in relation to MYC and mitotic stress [22] and that inhibition of apoptosis can similarly impair precancerous lymphoma in Eμ-Myc mice [36]. This suggests that these observations are likely more general to MYC oncogenes and require further investigation. Finally, while our combination therapy approach using MCL-1 inhibitors/barasertib combination therapy showed a significant extension of life in an animal model of neuroblastoma, these benefits were relatively modest. This suggests further exploration of dosing conditions, or identification of optimum anti-mitotic/pro-apoptotic agents is required, for instance using biologically relevant testing platforms such as orthotopic patient-derived xenograft models for neuroblastoma and potentially other MYC(N) driven cancers. Moreover, biomarkers for treatment personalization should be explored, with MYCN amplified neuroblastoma a logical patient population that may benefit. With optimized combination therapies devised, these approaches present potential new relapse-targeted therapies in a personalized medicine context.

## 4. Materials and Methods

### 4.1. Neuroblastoma Cell Line Tissue Culture

Kelly and SK-N-AS were obtained from the European Collection of Cell Cultures. SK-N-BE(2)C and SH-EP, were provided by Dr June Biedler, Memorial Sloan-Kettering Cancer Centre, New York. Kelly was cultured in Roswell Park Memorial Institute (RPMI) 1640 with 10% FCS (Life Technologies, Carlsbad, CA, USA). The remaining cell lines were cultured in Dulbecco’s modified Eagle’s medium (DMEM) with 10% FCS. SHEP21N cells were induced with 2 μg/mL doxycycline or DMSO of equal volume for 24 h, prior to drug treatment or siRNA transfection. The identity of each cell line was verified by short tandem repeat genetic profiling (CellBank Australia, Sydney, Australia).

### 4.2. siRNA Transfection

Cells were reverse transfected with 40 nM siRNA and Lipofectamine 2000 under serum-free condition for 6 h followed by culturing overnight in a normal growth medium. siRNAs used were SMARTpool siRNA (Dharmacon, Ste 100, Lafayette, CO, USA) AURKB siRNA (M-003326-08), SMARTpool BUB1B siRNA (M-004101-02), and SMARTpool KIFC1 siRNA (M-004958-02). On-target plus control siRNA (001810-10-20) was used as control.

### 4.3. Cell Viability Assays

For drug treatment, cells were plated in 96-well plates and treated the following day with various compounds. For siRNA transfection, cells were reversely transfected in 96 well plates. Viability was determined after drug treatment or transfection using a resazurin-based assay (654.5 μM resazurin, 78.2 μM methylene blue, 1 mM potassium hexacyanoferrate (III), 1 mM potassium hexacyanoferrate (II) trihydrate in phosphate-buffered saline), with fluorescence (Ex/Em 530–560/590 nm) measured using a Victor 3 multilabel plate reader (PerkinElmer, Macquarie Park, Australia). Data displayed are the mean viability normalized to control cells ± standard error of the combined data from three independent biological replicates.

### 4.4. Flow Cytometry Assays

Cell cycle analysis and apoptosis assay were performed on transfected or drug-treated cells. For apoptosis assay by flow cytometry, cells were stained with PE-Annexin-V and 7-AAD (559763, BD Pharmingen) according to the manufacturer’s instructions. Samples were analyzed using the FACSCalibur (BD Biosciences, Macquarie Park, NSW, Australia) and the data was analyzed using the FlowJo software v10.9 (Ashland, OR, USA). Viable cells were negative for both Annexin V and 7-AAD staining. Cells undergoing apoptosis were Annexin V positive. Cells in the late apoptosis stage or necrosis were also positive for 7-AAD staining. For cell cycle analysis, cells were fixed in ice-cold 70% ethanol and stained with Propidium Iodide (PI) (556463, BD Pharmingen) as previously described [42]. DNA content was measured at 670 nm by flow cytometry. Data analysis was carried out using the Flowjo built-in Cell Cycle Analysis platform.

### 4.5. Colony Formation Assays

Cells were transfected with siRNA in the presence of Lipofectamine 2000 24 h prior to being seeded in 6-well plates (200 cells/well for SH-EP, 250 cells/well for SK-N-BE(2)C and SK-N-AS). Once colonies had formed (7 days for SK-N-BE(2)C, 9 days for SK-N-AS and SH-EP), cells were fixed with cold methanol and stained with 0.5% crystal violet in 50% methanol. Colonies were counted using ImageJ software v1.54g. The data presented are the mean colony number normalized to control ± error of the combined data from three independent biological replicates.

### 4.6. Western Blotting

Protein was extracted from whole cell lysate in RIPA buffer (150 mM NaCl, 1.0% IGEPAL CA-630, 0.5% sodium deoxycholate, 0.1% SDS, 50 mM Tris, pH 8.0; Sigma-Aldrich, Macquarie Park, Australia) containing 1% Triton X-100 detergent (Sigma-Aldrich) with 10% protease inhibitor (Sigma-Aldrich, Macquarie Park, Australia), followed by sonication (UNIFXP12, Unisonic) for 15 min at 4 degrees Celsius. Protein concentration was determined using the Pierce BCA (bicinchoninic acid) Protein Analysis Kit (Pierce) as per the manufacturer’s instructions. Before electrophoresis, protein was denatured at 95 degree for 5 min in loading buffer with reducing agent (58 mM Tris, 100 mM DTT, 58 mM SDS, 30%*v/v* glycerol, bromophenol blue), and 30 ug protein was electrophoresed on 10% TGX precast gel (Bio-Rad, South Granville, Australia Cat# 5671033) and transferred to a nitrocellulose membrane (Bio-Rad, Cat# 1620112). Membranes were blocked in 5% skim milk in Tris-buffered saline for 1 h at room temperature, and incubated with various primary antibodies in Tris-buffered saline with 0.5% Tween-20 at 4 degrees overnight (Rabbit anti-AuroraB, 1:1000, Abcam, Cambridge, UK, Cat# ab-2254; rabbit anti-BUB1b, 1:1000, Proteintech Cat# 11504-2-AP; rabbit anti-KIFC1, 1:1000, cell signaling Cat# 12313S, mouse anti-MYCN, 1:1000, Santa-Cruz Biotechnology, Dallas, Texas, USA, Cat# sc-53993; rabbit anti-PARP, 1:1000, Cell signaling, Cat# 9542S). After washing, membranes were incubated in anti-mouse or anti-rabbit horseradish peroxidase antibodies (1:3000, Life Technologies, Cat# 31430, 31460) for 1 h at room temperature. Immunocomplexes were visualized by Clarity ECL (Bio-Rad, South Granville, Australia Cat# 1705061) and ChemiDoc MP Imaging System (Bio-Rad, South Granville, Australia). GAPDH (Mouse anti-GAPDH. 1:10,000, Santa-Cruz Biotechnology, Dallas, Texas, USA, Cat# sc-365062) was used as loading control as indicated.

### 4.7. Drug Treatments In Vitro

Drugs used were sourced as follows: vincristine (Selleckchem, S1241), barasertib (Selleckchem, S1147), VX-680 (Selleckchem, S1048), ABT199 (Selleckchem, S8048), S63845 (Selleckchem, S8383), ABT-263 (Selleckchem, S1001). Drug combinations were applied as a constant ratio drug series. The area under the curve (AUC) was calculated using GraphPad Prism 9 using log-10 adjusted dose. AUC is represented as the percentage of the curve with the maximum area (i.e., the least potent condition). Bliss additivity was calculated as described [33] for each dose combination and a theoretical drug additivity curve was inferred from these data points. Statistical difference between AUCs was evaluated using the method described by GraphPad Prism 9. Synergy was calculated as the difference between the combination AUC and the additive AUC calculated using the Bliss method described above. The potency of a combination was calculated as 1 minus the combination AUC.

### 4.8. Combination Therapy In Vivo

TH-MYCN^+/+^ mice were monitored for tumor growth by palpation since weaning. Mice were assigned to four treatment groups: vehicle (saline), intraperitoneal barasertib (Selleck, Cat# A-1377), intravenous S63845 (Active Biochem, Cat# A-6044), or the combination of barasertib and S63845. The treatment scheme for barasertib was 50 mg/kg/day, 4 consecutive days per week for two weeks, starting from the detection of a 2–3 mm tumor. The treatment scheme for S63845 was 25 mg/kg/day for 5 consecutive days starting from one week after the detection of 2 to 3 mm tumors. Barasertib was dissolved in 30% PEG400, 0.5% Tween 80, and 5% Propylene glycol. S63845 was dissolved in 25 mM HCl, 20% 2-hydroxy propyl β-cyclo dextrin. The mice were monitored continuously for tumor growth by palpation since the start of treatment and were humanly sacrificed when the tumor reached 10 mm in diameter. The Kaplan-Meier plot was constructed to assess the effect of single agents and combination therapy. Statistical significance was determined using log-rank tests.

### 4.9. Prophylactic Treatment in TH-MYCN Mice with Neuroblast Hyperplasia

Six-day-old TH-MYCN^+/+^ mice were treated intraperitoneally with vincristine (0.05 mg/kg/day in 5% dextrose), barasertib (25 mg/kg/day in 30% PEG400, 0.5% Tween 80 and 5% Propylene glycol), ABT263 (100 mg/kg/day in 5% DMSO, 95% corn oil) or saline for 4 consecutive days. Mice were humanely sacrificed at 12 days old and fixed in 10% neutral buffered formalin for 24 h followed by 80% ethanol for 5 days. Skin, teeth, and limbs were removed, and mice were bisected longitudinally. Fixed tissues were paraffin-embedded and then serial sections were cut to identify the location of any ganglia by H&E staining. All mice were examined for the hyperplastic ganglia by two examiners independently. Representative photos were taken using an BX53 light microscope (Olympus, Shinjuku-ku, Tokyo, Japan) and DP-73 camera with cellSens software v4.2 (Olympus, Shinjuku-ku, Tokyo, Japan). Data are presented as the percentage area of hyperplasia ± standard deviation. For examination of tumor incidence and growth, prophylactic treatment proceeded on a 4-day-on/3-day-off schedule for a total of 22 doses. Tumor growth was monitored by palpation and the mice were humanely sacrificed when the tumors reached 10 mm in diameter, or the experiment continued for 210 days, whichever came first. Statistical significance was determined using log-rank tests.

### 4.10. Single-Cell Quantitative PCR

Sympathetic ganglia from 10-day-old TH-MYCN^+/+^ mice or tumors from 40-day-old TH-MYCN^+/+^ mice were dissected and single cells were dissociated using a trypsin/collagenase-based method as previously described [7,15]. Viability was confirmed to be >90% using Trypan blue staining. Single-cell qPCR was undertaken using Fluidigm C1 for targeted qPCR. The following targeted Taqman probes (Thermo Fisher Scientific, Waltham, MA, USA) were used for the qPCR.


**Gene**

**Probe**

**Gene**

**Probe**
Phox2bMm00435872_m1Bub1bMm00437811_m1hMYCNHs00232074_m1Kif23Mm00458527_m1Dlk1Mm00494477_m1Bub1Mm00660135_m1Gap43Mm00500404_m1Kifc1Mm00835842_g1DbhMm00460472_m1AspmMm00486659_m1Tubb3Mm00727586_s1Plk4Mm00550358_m1Sox10Mm01300162_m1Depdc1aMm01319324_g1Col1a2Mm00483888_m1Ccnb2Mm01171453_m1S100a1Mm01222827_m1Prr11Mm00723607_m1

### 4.11. Bioinformatics Analysis

All bioinformatic analyses were conducted using R software v4.02.

Raw microarray data from bulk ganglia are available in the ArrayExpress database under accession number E-MTAB-3247. Microarray data was processed and expression change was quantified using average expression per genotype and time points using a multiple linear model as described previously [15]. Heatmap visualization of data was undertaken using the ComplexHeatmap package [43] using ward. d2-based hierarchical clustering. Gene ontology overrepresentation was undertaken using clusterProfiler package [44] showing representative GO terms for each gene group. Scatter plots were developed using the ggplot2 package [45].

Single-cell qPCR data was processed using the Singular analysis toolset v3.52 (Fluidigm) using a default limit of detection (LOD) of 24. Expression data were normalized to log2 space using LOD—cycle-threshold (Ct) for each gene. Signatures for respective cell type markers were calculated as the average Z-score in all cells for those genes. To classify cell type, the following signature thresholds were used: Schwannian sig > 0.75 (Schwannian cells), Neuroblast sig > 0 & Neuroblast sig > Ganglion sig (Neuroblast), Ganglion sig > 0 and Ganglion sig > Neuroblast sig (Ganglion cell). All other cells below these thresholds were removed from the analysis. Heatmap visualization of data was undertaken using the ComplexHeatmap package [43]. Plotting of box plots was undertaken using the ggpubr package [46]. Scatter plots were developed using the ggplot2 package [45].

Survival data were imported and processed using the survival package [47]. Plotting and log-rank *p* values were calculated using method “1” using the survminer package [48].

Neuroblastoma tumor microarray for 475 patients [24] was obtained from gene expression omnibus (GEO) under accession GSE45480. Heatmap visualization of data was undertaken using the ComplexHeatmap package [43] using ward.d2-based hierarchical clustering.

ChIP-seq (GSE80151) raw fastq files were obtained directly from the European Nucleotide Archive (ENA) under the study accession PRJNA318044 [29]. Reads from the fastq files were first quality trimmed using trimgalore, followed by alignment to the human genome (GRCh38) using bowtie2 [49]. SAMtools was then used to convert, sort, and index alignments [50]. Peaks were then called using MACS2 either in single-end mode, with an FDR q-value threshold of <0.05 [51]. Fold enrichment tracks which represent the relative enrichment of the ChIPed protein compared to the genomic input, were generated using MACS2 and converted to the bigwig format using BEDtools for visualization [52].

### 4.12. Statistical Analysis

Unless otherwise stated, all statistical analysis was conducted in GraphPad Prism software v9, and *p*-values were determined using unpaired two-sided *t*-tests. Correlation analysis represents Pearson statistics. Throughout the manuscript *, *p* < 0.05, **, *p* < 0.01, ***, *p* < 0.001, ns, non-significant.

## 5. Conclusions

In conclusion, we identified early upregulation and dependence on mitotic dysregulation in MYCN-driven tumorigenesis. This molecular dependence is a vulnerability that could be leveraged for better combination therapy efficacy using an antimitotic/pro-apoptotic strategy.

## Figures and Tables

**Figure 1 ijms-24-15571-f001:**
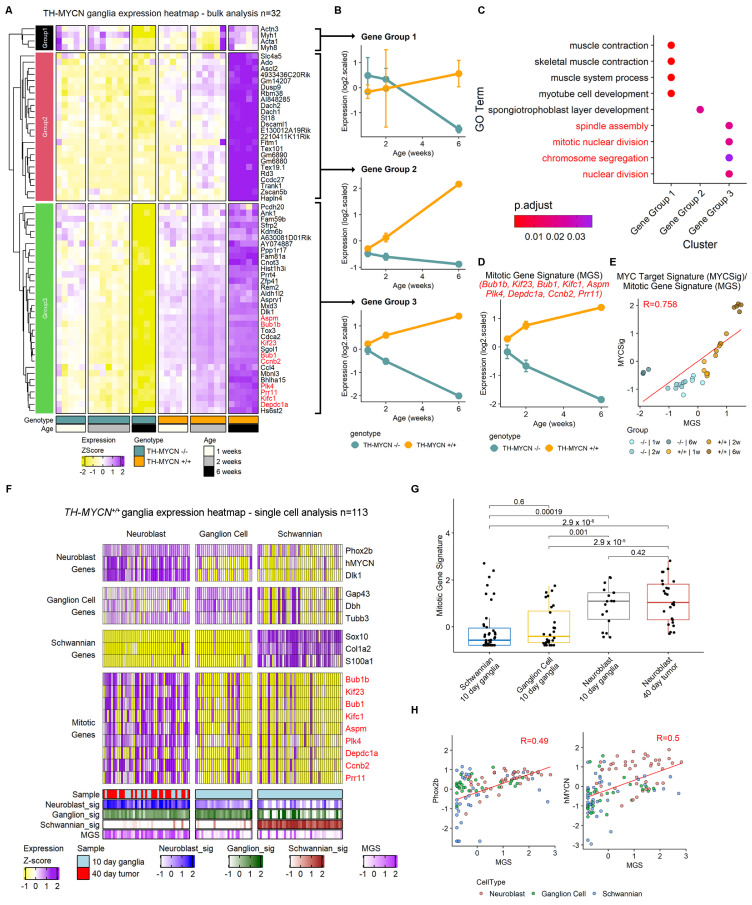
Pre-malignant neuroblasts in *TH-MYCN^+/+^* sympathetic ganglia show mitotic dysregulation. (**A**) Heatmap showing microarray gene expression patterns for wild-type and homozygous *TH-MYCN* ganglia and tumors. The genes shown were identified using a multiple linear model [15]. Ward.d2-based hierarchical clustering identified three gene groups. Red print indicates genes that are used in the mitotic gene signature (MGS). (**B**) Scatter plots showing average expression for gene signatures created from gene groups identified in (**A**). Signatures were created using average Z-scores of all genes in the signature. Error bars represent standard deviation. (**C**) Gene ontology (GO) term enrichment using genes from each gene group identified in (**A**,**B**). Red print corresponds to GO terms related to mitosis (enriched in Gene Group 3). Representative GO terms were selected, see the full results analysis in Appendix A. Color represents adjusted *p*-value. (**D**) Scatter plots showing MGS: the mitotic genes identified in Gene group 3. Signatures were created using average Z-scores of all genes in the signature. Error bars represent standard deviation. (**E**) Correlation comparing MGS and a MYC-target gene signature [16] expression for *TH-MYCN* ganglia and tumors. R-value refers to Pearson correlation statistic. Color corresponds to genotype/age groupings. (**F**) Heatmap showing single-cell qPCR gene expression patterns for *TH-MYCN^+/+^* ganglia (10 days—light blue) and tumors (40 days—red). Cell type classifications were made using the expression patterns of neuroblast, ganglion, and Schwannian signatures. Genes in red print correspond to mitotic genes in the MGS. (**G**) Boxplot comparisons of MGS expression in different cell and sample types. *p*-values derived from the Wilcoxon rank sum test. (**H**) Correlation comparing MGS and neuroblast markers Phox2b (**left**) and hMYCN (**right**) for expression single cells from TH-MYCN^+/+^ ganglia and tumors. R-value refers to the Pearson correlation statistic. Color corresponds to cell type groupings.

**Figure 2 ijms-24-15571-f002:**
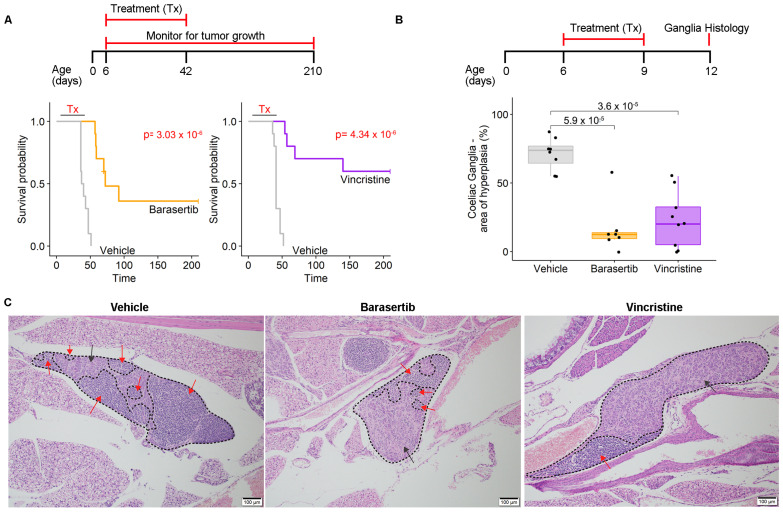
Prophylaxis with antimitotic compounds impairs neuroblastoma tumorigenesis in TH-MYCN^+/+^ mice. (**A**) Kaplan–Meier curve for *TH-MYCN^+/+^* mice survival assessing vehicle vs. 25 mg/kg/day barasertib (**left**) and 0.05 mg/kg/day vincristine (**right**). Treatment was from day 6 until day 42 on a 4-day-on/3-day-off schedule for a total of 22 doses. The endpoint was considered to be time until the maximum palpable tumor was detected (10 mm diameter) or 210 days, whichever came first. *p*-value comparing treatments was calculated using log-rank tests. (**B**) Boxplot comparing different treatment groups for percentage of hyperplasia in the coeliac ganglia of treated mice: vehicle vs. 25 mg/kg/day barasertib and 0.05 mg/kg/day vincristine. Treatment was from day 6 until day 9 for a total of 4 doses. Ganglia were detected in hematoxylin and eosin sections of mice at 12 days of age. (**C**) Representative hematoxylin and eosin sections from (**B**) shows hyperplastic (red arrows) and non-hyperplastic (black arrows) regions in the coeliac ganglia.

**Figure 3 ijms-24-15571-f003:**
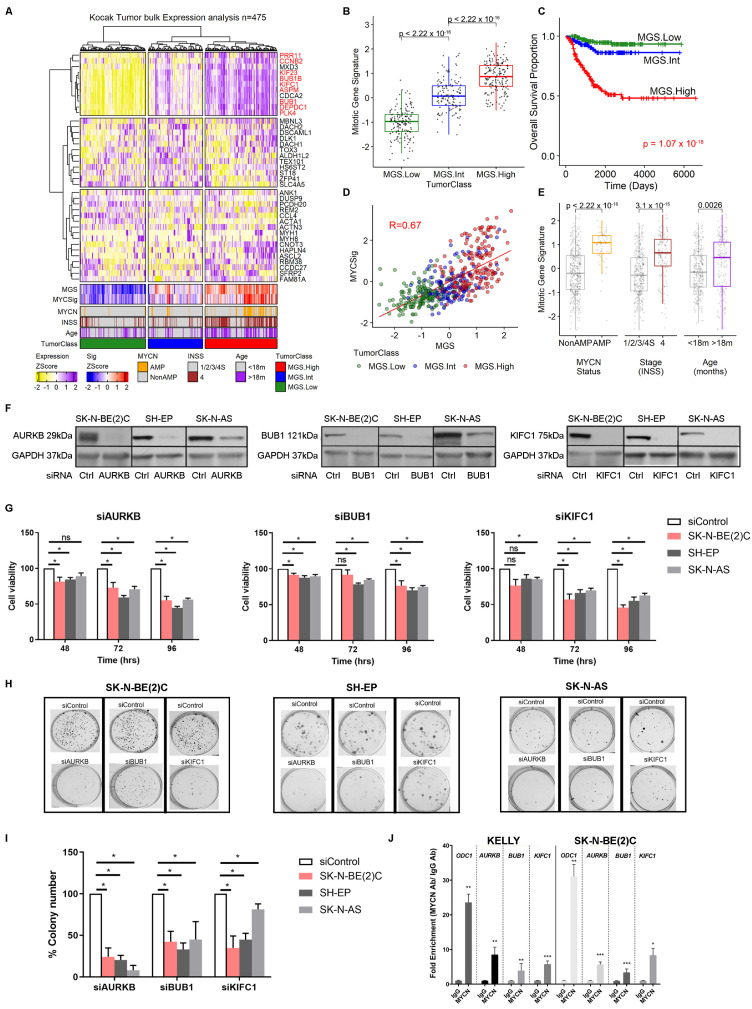
Mitotic dysregulation occurs in established neuroblastoma and is essential for cell viability and proliferation. (**A**) Heatmap from neuroblastoma tumor microarray [24] for the same genes identified in Figure 1A. Ward.d2-based hierarchical clustering identified three tumor groups (see TumorClass). Red print genes indicate human genes that constitute the MGS. (**B**) Boxplot showing MGS expression in tumor microarray split by tumor grouping in (**A**). *p*-values derived from the Wilcoxon rank sum test. (**C**) Kaplan–Meier plot for overall survival, split based on tumor groupings identified for (**A**,**B**). *p*-value comparing treatments was calculated using log-rank tests. (**D**) Correlation comparing MGS and a MYC-target gene signature [16] expression in neuroblastoma tumors. R-value refers to the Pearson correlation statistic. Color corresponds to tumor groupings. (**E**) Boxplot showing MGS expression in neuroblastoma tumors split by clinical groups for MYCN status, Stage, and Age. *p*-values derived from the Wilcoxon rank sum test. (**F**) Western blot for neuroblastoma cell lines treated with select mitotic gene knockdown, using pools of control or targeted siRNA. (**G**) Resazurin-based assays showing cell viability relative to control siRNA-treated cell lines. Data shows average expression from three biological replicates. Error bars relate to standard error. *p* values were calculated using a *t*-test. *, *p*-value < 0.05, ns, non-significant. (**H**) Representative images of colony assays for neuroblastoma cell lines treated with different siRNA conditions. (**I**) Colony assays showing colony number relative to control siRNA-treated cell lines. The data show average expression from three biological replicates. Error bars relate to standard error. *p* values were calculated using a *t*-test. *, *p* value < 0.05. (**J**) Chromatin immunoprecipitation PCR assays testing MYCN enrichment in the promoter region of miotic genes. *ODC1* promoter was used as a positive control. IgG used a control antibody for non-specific enrichment. *, *p* value < 0.05, **, *p*-value < 0.01, ***, *p*-value < 0.001.

**Figure 4 ijms-24-15571-f004:**
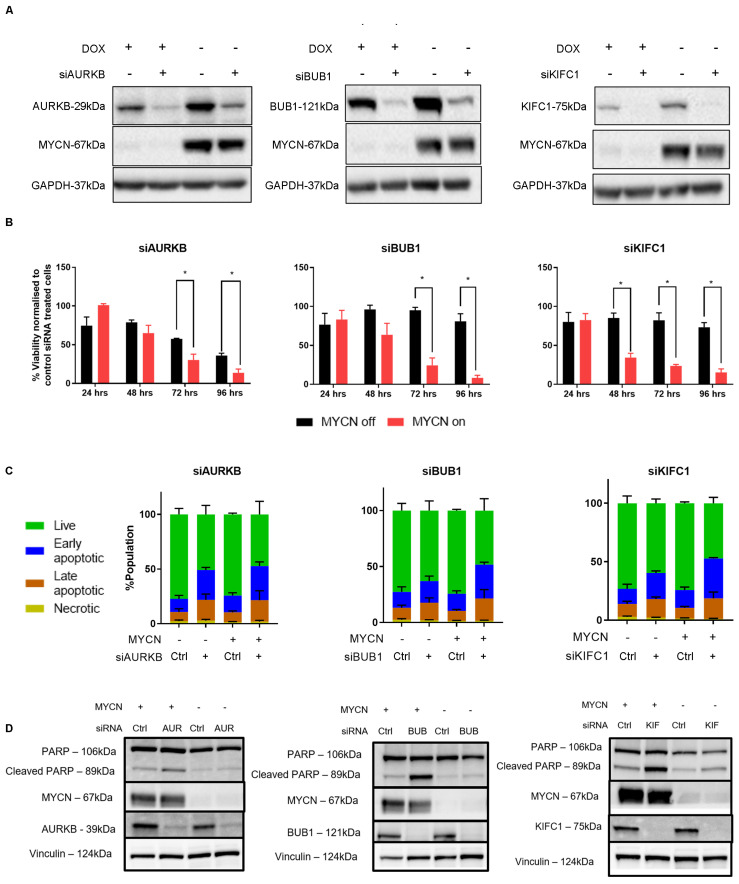
Genetic inhibition of mitosis leads to selective apoptosis in neuroblastoma cells with MYCN overexpression. (**A**) Western blot for mitotic gene and MYCN expression in SHEP21N cells treated with variable conditions for MYCN induction (DOX = MYCN-ve, DMSO = MYCN+ve) and mitotic siRNA. (**B**) Resazurin-based assays showing cell viability relative to control siRNA-treated SHEP21N cells with variable MYCN expression. Data shows average expression from three biological replicates. Error bars relate to standard error. *p* values were calculated using a *t*-test. *, *p*-value < 0.05. (**C**) Annexin V/7-AAD assays to classify the proportion of apoptotic/necrotic cells in SHEP21N cells treated with variable mitotic siRNA or MYCN induction. (**D**) Western blots for apoptotic marker cleaved PARP treated with variable conditions for MYCN induction and mitotic siRNA.

**Figure 5 ijms-24-15571-f005:**
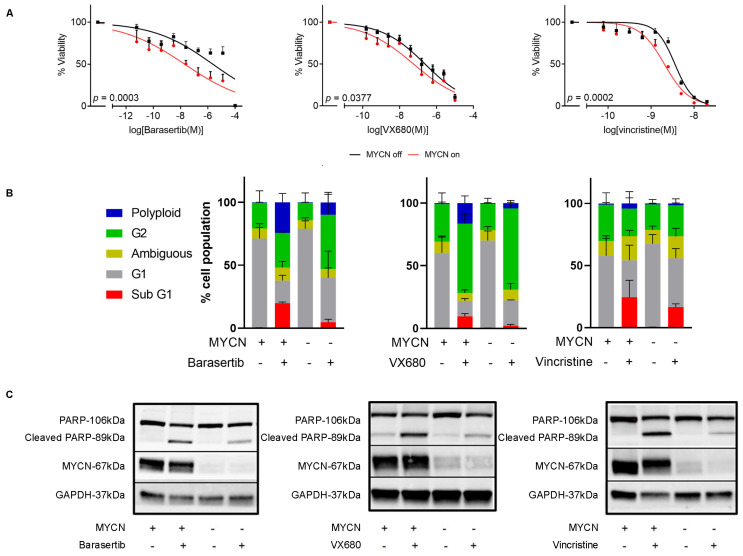
Chemical inhibition of mitosis leads to selective apoptosis in neuroblastoma cells with MYCN overexpression. (**A**) Resazurin-based assays showing cell viability antimitotic dose response relative to untreated SHEP21N cells with variable MYCN expression. Data shows average expression from three biological replicates. Error bars relate to standard error. *p* values were calculated based on IC50 value using extra-sum of squares F-Test. (**B**) Cell cycle assays using propidium iodide to classify the proportion of cell cycle stages for cells in SHEP21N cells treated with variable antimitotic compounds or MYCN induction. (**C**) Western blots for apoptotic marker cleaved PARP treated with variable conditions for MYCN induction and antimitotic compounds.

**Figure 6 ijms-24-15571-f006:**
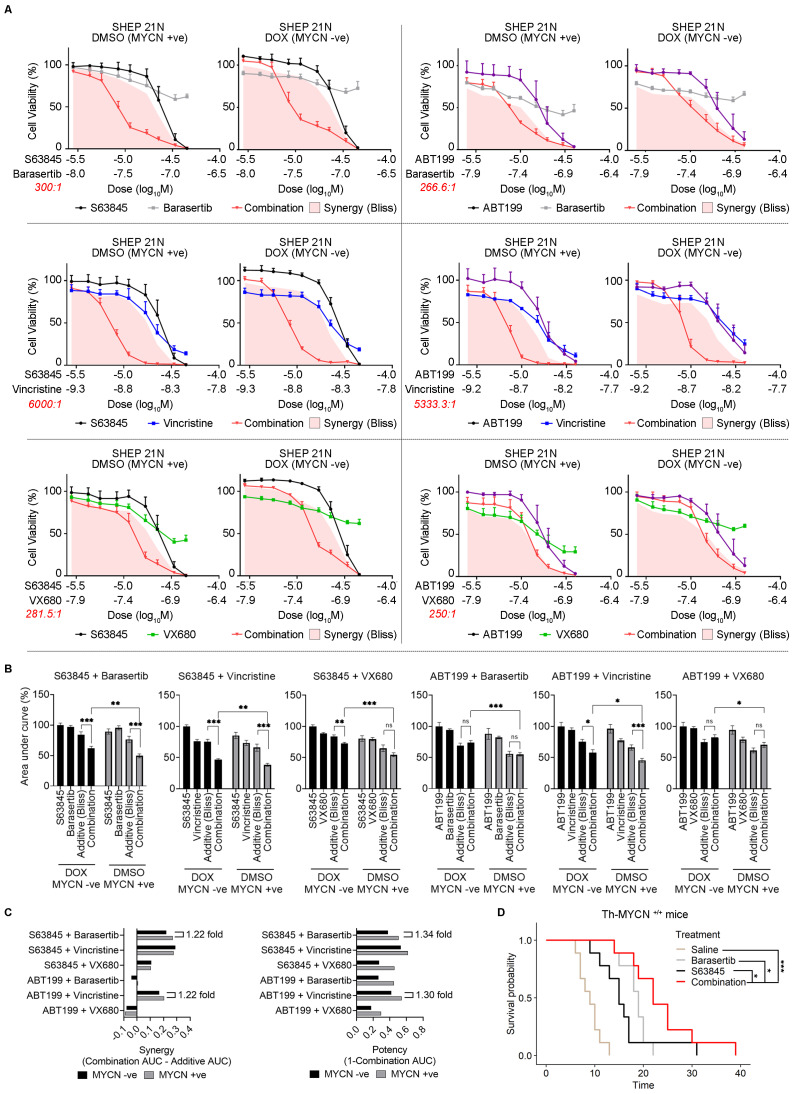
Antimitotic/pro-apoptotic combination therapy is more effective in the presence of MYCN overexpression. (**A**) Resazurin-based combination therapy assays comparing constant ratio dose series of antimitotic and pro-apoptotic compounds in SHEP21N cells under differing MYCN induction. The theoretical additivity of each dose combination was calculated according to the Bliss equation [33]. Any region exceeding Bliss additivity is considered to be synergism and is interpolated on the chart. (**B**) Area under the curve calculations for each combination therapy assay is shown in (**A**). AUC was calculated relative to the least potent single agent across both MYCN+ve and MYCN-ve conditions. *p*-value calculated using a *t*-test. *, *p* value < 0.05, **, *p*-value < 0.01, ***, *p*-value < 0.001, ns, non-significant. (**C**) Synergy (**left**) and potency (**right**) were calculated based on a combination of AUC as shown. These metrics are shown for each combination therapy tested, showing a comparison between MYCN-ve and MYCN+ve conditions. (**D**) Kaplan Meier curve for TH-MYCN^+/+^ mice survival assessing vehicle, 50 mg/kg/day single-agent barasertib, 25 mg/kg/day single-agent S63845 and combination barasertib/S63845 (25 and 50 mg/kg/day respectively). Treatment was when established tumors were detected. See Section 3 for treatment schedules. The endpoint was considered to be time until the maximum palpable tumor was detected (10 mm diameter). *p*-value comparing treatments was calculated using log-rank tests. *, *p* value < 0.05, ***, *p*-value < 0.001.

## Data Availability

All data generated in this article is available upon request to the corresponding author.

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
