# Peer review of "Mitotic Dysregulation at Tumor Initiation Creates a Therapeutic Vulnerability to Combination Anti-Mitotic and Pro-Apoptotic Agents for MYCN-Driven Neuroblastoma"

_ijms, 2023, doi:10.3390/ijms242115571_

Round 1

Reviewer 1 Report

The molecular pathways involved in MYCN-driven neuroblastoma initiation are not completely deciphered. In this manuscript, the authors conducted a groundbreaking study to unravel the molecular mechanisms of tumor initiation in MYCN-driven neuroblastoma. Using downregulation and overexpression of mitotic relevant components as well as treatment with antimitotic and pro-apoptotic drugs, the authors demonstrated that the mitotic dysregulation is a main component in tumor initiation in early neuroblasts that can be inhibited by the combination of antimitotic compounds and pro-apoptotic compounds in MYCN-driven neuroblastoma
Moreover, this article showcases a well-structured and organized format, encompassing all the anticipated data. Considering the importance of this work, we recommend accepting it for publication in IJMS.

Author Response

Thank you very much to the reviewer for their time and effort in review of our article!

Reviewer 2 Report

In the present work Zhai et al. reported on the role of mitotic dysregulation at tumor initiation in MYCN-driven neuroblastoma. The authors reported that mitotic dysregulation is significant for the progression of early neuroblastoma tumor initiation and that a combination of antimitotic compounds and pro-apoptotic compounds offers synergistic therapeutic benefit in MYCN-driven neuroblastoma. Their work is very interesting and I have enjoyed reading the present manuscript. It is well-written and the topic is interesting. The present work has merit for publication. 

A point that I would like to clarify; the authors suggest that mitotic dysregulation makes neuroblastoma cells prone to cell death at early onset. Thus, do the authors suggest that if neuroblastoma is diagnosed at its early stages (maybe before clinical presentation?) therapy is more probable to have a better effect? Please comment on this. 

It is known that tumors manifest cell cycle dysregulation, why was MYCN significant in this process? How the authors did connect the role of MYCN to mitotic dysregulation? 

Mention the limitations of the study. What can be done better, or what are the next steps? How can treatment be modified for more efficient disease remission? The authors should highlight their results and mention how their findings could prove useful for the treatment of neuroblastoma.

Overall, the authors have done a great deal of effort and experimentation, which is to their appraisal and furthermore, they have done an excellent work presenting their hypothesis.

Reviewer 3 Report

General Impressions

This study offers a detailed exploration of mitotic dysregulation in the context of MYCN-driven neuroblastoma, leveraging the TH-MYCN transgenic mouse model. Overall, the study appears to be comprehensive, built on a robust experimental framework, and provides significant insights into the therapeutic potential of targeting mitotic dysregulation in neuroblastoma.

Strengths

In-depth analysis: The manuscript provides a thorough analysis, particularly through bulk transcriptomics and comparative single cell qPCR, to showcase the molecular events tied to tumor initiation and progression.

Significance of findings: The discovery of early mitotic dysregulation in neuroblastoma tumorigenesis is of notable significance. This could have potential implications for the therapeutic targeting of MYCN-driven neuroblastomas.

Translational potential: The efficacy of prophylactic treatment with antimitotic agents and the synergistic effects of combination therapy provide promising avenues for future clinical exploration.

Areas for Improvement

Editorial suggestions: Please place the 'Materials and Methods' section before the 'Discussion' section. There should be a separate chapter for 'Conclusion'."

Clinical Implications: While the therapeutic potential has been explored in depth, a further discussion on how these findings can be translated to clinical settings and what challenges might be encountered would add depth.

Comparative Analysis: It might be interesting to see how MYCN-driven neuroblastomas compare with other tumor types in terms of mitotic dysregulation and therapeutic vulnerabilities.

Broader Discussion: The manuscript might benefit from a broader discussion on the role of MYCN in other cancers or diseases, helping to situate the findings in a larger context.

Clarity and Coherence: The language used is clear and coherent, suitable for the intended scientific audience. The paper effectively communicates complex scientific concepts and experimental findings.

Vocabulary: The vocabulary is advanced and field-specific, which is typical for scientific literature, especially in the domain of cellular biology and oncology.

Sentence Structure: The sentences are structured appropriately, using a combination of compound and complex sentence structures to convey the information. While some sentences are lengthy, they are still comprehensible, given the nature of scientific writing.

Grammar and Usage: There are no evident grammatical errors in the provided text.

Flow and Continuity: The paper follows a logical flow, starting with an introduction that provides context, followed by specific findings, and culminating in a discussion.

Paragraph Structure: Paragraphs are well-structured, each addressing specific aspects or results of the research.

In summary, the quality of the English language in this document is of a high standard, consistent with what one would expect from a professional, peer-reviewed scientific journal article. The content is detailed and complex, but the language effectively conveys the intended research findings and concepts.
